# Distributed Assessment of Virtual Insulin-Pump Settings Using SmartCGMS and DMMS.R for Diabetes Treatment

**DOI:** 10.3390/s22239445

**Published:** 2022-12-02

**Authors:** Martin Ubl, Tomas Koutny, Antonio Della Cioppa, Ivanoe De Falco, Ernesto Tarantino, Umberto Scafuri

**Affiliations:** 1Department of Computer Science and Engineering, University of West Bohemia, Technicka 18, 330 01 Pilsen, Czech Republic; 2Department of Computer Science and Engineering, New Technologies for Information Society, University of West Bohemia, Technicka 18, 330 01 Pilsen, Czech Republic; 3Natural Computation Lab, Department of Information Engineering, Electrical Engineering and Applied Mathematics, University of Salerno, Via Giovanni Paolo II 132, 84084 Fisciano, Italy; 4ICAR-National Research Council of Italy, Via P. Castellino, 80131 Naples, Italy

**Keywords:** diabetes, insulin pump, controller, in silico, smartcgms

## Abstract

Diabetes is a heterogeneous group of diseases that share a common trait of elevated blood glucose levels. Insulin lowers this level by promoting glucose utilization, thus avoiding short- and long-term organ damage due to the elevated blood glucose level. A patient with diabetes uses an insulin pump to dose insulin. The pump uses a controller to compute and dose the correct amount of insulin to keep blood glucose levels in a safe range. Insulin-pump controller development is an ongoing process aiming at fully closed-loop control. Controllers entering the market must be evaluated for safety. We propose an evaluation method that exploits an FDA-approved diabetic patient simulator. The method evaluates a Cartesian product of individual insulin-pump parameters with a fine degree of granularity. As this is a computationally intensive task, the simulator executes on a distributed cluster. We identify safe and risky combinations of insulin-pump parameter settings by applying the binomial model and decision tree to this product. As a result, we obtain a tool for insulin-pump settings and controller safety assessment. In this paper, we demonstrate the tool with the Low-Glucose Suspend and OpenAPS controllers. For average ± standard deviation, LGS and OpenAPS exhibited 1.7 ± 0.6% and 3.2 ± 1.8% of local extrema (i.e., good insulin-pump settings) out of all the entire Cartesian products, respectively. A continuous region around the best-discovered settings (i.e., the global extremum) of the insulin-pump settings spread across 4.0 ± 1.1% and 4.1 ± 1.3% of the Cartesian products, respectively.

## 1. Introduction

Diabetes mellitus (DM) is a disease of civilization characterized by elevated blood glucose (BG) level [1]. It is caused either by an absolute (type 1 diabetes, T1D) or a relative (type 2 diabetes, T2D) insulin insufficiency. Insulin is a hormone produced by the pancreatic β-cells. It promotes glucose utilization in cells, thus lowering BG. Additionally, it stimulates glycogenesis—the process of transforming glucose to glycogen. Glycogen is a storage form of glucose, which comprises many glucose molecules.

The acute complications of DM are hypo- and hyperglycemia. The latter is caused by insufficient insulin in blood plasma, while the former is caused by the exact opposite [2]. Chronic complications, which are generated by severe hyperglycemic episodes, often include organ damage that may eventually lead to patient death. We often refer to an ideal glucose range of 3.5–7.5 mmol/L as normoglycemia [1,2]. This range varies with age and associated condition [3].

Regarding T1D, to avoid short- and long-term complications, insulin is dosed externally [4]. The insulin pump can deliver insulin automatically. Insulin pump controllers exist, each with different parameters and internal logic. In this study, we propose an evaluation method that uses an FDA-approved diabetic patient simulator to verify the pump’s safety before it enters the market.

### 1.1. Insulin Pump

Let us consider a patient with an insulin pump. Such a patient continuously measures interstitial glucose levels using a continuous glucose monitoring system (CGMS). CGMS measures the level in the interstitium. A physiological lag between the interstitium and BG is induced by a delayed, facilitated diffusion between the blood and interstitium compartments [5,6]. In addition, there is a technological lag caused by CGMS sensor construction and its internal model of measured-signal processing. CGMS estimates true BG from the measured signal (IG). When using modern-day sensors, the BG and IG values do not exhibit any significant difference and it is approved to use IG values only for glycemic control [7].

To control glucose levels, the patient uses an insulin pump—a device that doses insulin automatically based on its settings [8]. The pump operates in two modes—bolus and basal insulin dosage. The former is a one-time delivery of a greater insulin volume, often delivered in multiple small doses over a short period.

The latter is a continuous dosage of smaller volumes throughout the whole day. Continuous subcutaneous insulin infusion (CSII) is achieved by combining these two modes.

A modern pump embeds a controller that reads IG from CGMS and adjusts the basal insulin delivery rate. We refer to such control as sensor-augmented pump therapy (SAP). The simplest example of SAP therapy is the low-glucose suspend (LGS) controller [9]. It stops insulin delivery for a selected period (e.g., 30 min) once IG drops below a selected threshold (e.g., 4 mmol/L) to avoid hypoglycemia. If the controller uses the predicted IG in addition to the current IG, we refer to such a controller method as predictive low-glucose management (PLGM) [10].

In addition, the controller can increase the basal insulin delivery rate to deal with potential or ongoing hyperglycemic episodes. Examples include various proportional(-integrative)-derivative (PD or PID) controllers [11,12], model-predictive control-based (MPC) algorithms [13,14], fuzzy logic controllers [15], and others.

Every controller must be evaluated for safety before entering the market [16]. Extensive in silico testing and verification must precede in vivo trials to avoid any harm to living subjects. Therefore, an in silico virtual patient and simulation environment is required. The controller must be evaluated within a certified in silico environment to provide the correct results. The Epsilon Group [17] developed the Diabetes Mellitus Metabolic Simulator for Research (DMMS.R) and its predecessor (T1DMS), which are to date the only simulators approved by the U.S Food and Drug Administration (FDA), including a cohort of virtual patients in trials. These simulators are approved for a single-day scenario with up to three meals daily [18].

The controller needs specific insulin-pump software. To evaluate the controller, we propose use of the SmartCGMS framework [19]. It is a signal-processing framework developed in modern C++, designed to run on low-power, safety-critical devices, such as an insulin pump. The controller can be integrated as a module and evaluated inside the SmartCGMS environment. Once it has passed verification tests successfully, it can be deployed to an insulin pump without changing its program code. This feature avoids any errors caused by differences between tested and target versions of the controller. DMMS.R enables evaluation of a limited subset of programming languages, even though most of them were not designed for embedded device operation (e.g., Matlab^®^ and JavaScript languages).

The latest versions of DMMS.R software support loading an external, compiled library that exports the controller from a native code. In addition, it can execute in a non-authoritative mode inside a native library, so the external code controls the simulation. We used this mode to connect DMMS.R with SmartCGMS. We describe this mechanism in Section 2.

The usual approach to in silico controller evaluation is to use a pre-defined simulation scenario (e.g., 24 h, three meals of 20, 40, and 60 g of carbohydrates, no insulin bolus) and manually find the best-suited controller parameters or use explicitly available parameters of the virtual patients. Then, the scenario is evaluated on the whole patient cohort and metrics are calculated. The usual evaluation metrics are:Time-in-range, i.e., percentage of measurements that lie in the normoglycemia range [20];Minimum and maximum error; possibly substituted by the 5th and 95th percentiles [21];Average error and standard deviation of individual errors.

For simplicity, the maximum and minimum levels are analyzed during the control variability grid analysis (CVGA) [21]. This method presents a grid that has the minimum (or the 5th percentile) value on the X-axis and the maximum (or the 95th percentile) on the Y-axis. For every patient, one point is placed into the grid. The grid is then divided into A, B, C, D, and E regions. Region A contains a feasible control outcome and region B contains a feasible control with benign deviations. Any other region contains infeasible control outcomes that may harm the patient by over-correction of hyper- or hypoglycemia or failure to deal with either.

Currently, there is a “do it yourself” (DIY) trend in CSII design, complementing certified products [22]. For example, the OpenAPS project [23] encourages its users to use a combination of non-certified software and legacy hardware. The use of old insulin pumps and sensors is required due to their communication protocol simplicity and transparency (with a risk of weak security [24,25]). The OpenAPS is distributed with a rule-based controller called *oref0*, although the recently improved version is often referred to as *oref1* [26].

### 1.2. SmartCGMS

SmartCGMS is a framework for signal analysis and processing [19]. It is written in modern C++ and is based on high-level architecture principles. It is designed to run on low-power devices with increased safety requirements. It comprises a set of entities. The top-level entity is called a *filter*. It serves a single isolated purpose, such as loading data, communicating with devices, such as sensors and pumps, stepping a mathematical model, calculating error metrics, saving a log to a file, rendering drawings, and many others. A filter may encapsulate other entities, such as discrete models, metrics, signals, and time-series approximators. A globally unique identifier (GUID) identifies every entity [27].

A configuration defines a filter set that is instantiated. Filters are connected in a linear chain. Entities communicate by passing small messages called *device events* [28]. A device event contains a single item of information passed from the event source filter through the chain to the last filter in the chain. Therefore, no filter can send events to any preceding filter unless it is a signal feedback filter. The feedback filter realizes a closed-loop control system inside the filter chain, with well-defined and constrained behavior to guarantee system stability. After a successful chain instantiation, the feedback sender filter is connected to a feedback receiver. During the operation, the feedback sender sends only a subset of the signal events through the feedback link to the receiver. For example, the feedback receiver could be an insulin pump driver where the loop is often closed through the patient and his/her CGM sensor.

A CSII controller could be implemented as a discrete model that reacts to all signals relevant to the controller (e.g., IG signal, consumed carbohydrates, dosed insulin, and others). The controller emits a new basal insulin delivery rate value when considered appropriate. The parameters of the controller are represented as parameters of the discrete model entity.

The SmartCGMS supports parameter optimization. Using a solver, it finds the best set of parameters for a given model, possibly encapsulated within a filter. For example, the meta-differential evolution [6] can be used with the crosswalk metric [29] to quantify model parameter fitness.

### 1.3. Ecosystem

A number of in silico metabolic models have been developed. The models differ in many respects, such as license, open-source vs. binary-only distribution, regulatory-body approval, patient cohort size and its diversity, drug simulation, user experience, and others. Table 1 summarizes the differences.

SmartCGMS can work with any metabolic model. The model can be implemented natively into SmartCGMS or it can be an external simulator. Therefore, the difference between the present work and the models lies in the ecosystem that we extend around the existing models. Specifically, we encapsulate each model with a SmartCGMS experimental setup. Hence, SmartCGMS provides a unified solver interface to determine and evaluate the parameter settings of each CSII controller. In addition, we deliver a unified user experience regardless of the underlying metabolic model.

## 2. Materials and Methods

The current CSII evaluation process relies on correctly estimating the controller parameters. When developing a controller based on physiological parameters as inputs, the parameters may be obtained directly from the simulator itself. Nonetheless, many controllers do not retain physiological parameters or their equivalents.

We propose a method to evaluate any controller using the SmartCGMS architecture in connection with the FDA-approved DMMS.R. As the controllers often do not follow a simple formula, contain rule-based exceptions, and have many other issues, the problem domain is usually not differentiable and may not be easily optimized to obtain the optimal parameters. For complex controllers, multiple parameter combinations may exhibit feasible control capabilities. The proposed method does not identify a single parameter set but identifies parameter regions with feasible control outcomes for a given patient.

### 2.1. Evaluation Method

To evaluate a controller, we either implement (LGS) or encapsulate (*oref0*) it as a SmartCGMS component, i.e., a discrete model. We configure the SmartCGMS to use this model, choose the parameters to evaluate, and estimate their lower (*L*) and upper (*U*) bounds and increments (*S*). Then, we calculate a vector *P* for each parameter *i* that contains all the values of a given parameter.
(1)$$P_{i}=\left\{L_{i},L_{i}+S_{i},L_{i}+2S_{i},...U_{i}\right\}$$

Then, we assemble an evaluation table *T* as a Cartesian product of all calculated vectors:(2)$$T=P_{1}\times P_{2}\times ...\times P_{n},$$
where *n* denotes the number of parameters to be evaluated.

To demonstrate the method, we chose n=2 and selected the two most significant parameters of all evaluated controllers, while setting the remaining parameters to a reasonable value for a given situation.

We demonstrated the method on the LGS and *oref0* (OpenAPS) controllers. For LGS, we evaluated the parameters of the default basal rate (*b*) and suspension threshold (Θ). The suspension period was fixed to 30 min. For *oref0*, we evaluated the default basal rate (*b*) and insulin sensitivity factor (*s*) parameters. We set the carbohydrate-insulin ratio to an optimal value provided by DMMS.R for each patient. Table 2 lists the parameter bounds and increments. The OpenAPS/*oref0* controller was configured with all its recent features and improvements (*oref1*, including unannounced meals, advanced meal assist, autotune, and super-micro boluses).

For each parameter combination in Table *T*, the controller was evaluated on all available virtual patients. Then, we gathered the metric value, which expresses the control quality of a given configuration.

Using these settings, we obtained 798 and 1645 metric values for LGS and *oref0*, respectively. As a controller target level, we used the average error metric value. We calculated the average metric value as an absolute difference between the actual IG and the constant value of 6.66mmol/L.

### 2.2. Distributed Evaluation Suite

To evaluate CSII, we need to execute a simulation for each element of the Cartesian product. Executing so many simulations is computationally intensive. Therefore, we propose a distributed evaluation suite comprising a SmartCGMS framework-based application, a network daemon, and DMMS.R.

We implemented the SmartCGMS front-end, which executes an evaluation scenario. The scenario, among other components, comprises a (testing-)scenario reader in the form of a log replay filter, a discrete model encapsulating a DMMS.R connection, a controller model, a feedback sender connected to the DMMS.R discrete model, and an error metric filter. Figure 1 shows the chain.

To allow distributed evaluation, we implemented a network daemon that forks an instance of DMMS.R to evaluate a single patient with a single parameter set. We implemented such a daemon to maintain a pool of DMMS.R workers to be used for parallel evaluation within isolated simulations, as depicted in Figure 2.

We use a TCP/IP network connection for offloading a part of the simulation to another network node. We implemented a specialized discrete model that establishes a connection to a remote node. During the initial handshake, a specific model and its virtual patient are selected. Instead of encapsulating the mathematical model into the discrete networking model, the model defers the simulation to a remote node, which executes the DMMS.R simulator instance. Within a single evaluation, one networked discrete model is directly associated with a single DMMS.R worker instance at the remote node. When a signal level is received, it is passed through the network to the remote network daemon, which accumulates all relevant levels (e.g., carbohydrate intake, bolus, and basal insulin). Once a simulation step is requested, it is also passed to the network daemon, which contacts the DMMS.R worker with all the accumulated signal levels and performs the actual stepping.

The network daemon component communicates with worker processes, such as events and mapped shared memory, using the local-only IPC. Signal levels are serialized into an array that is stored in the shared memory. Then, the daemon signalizes an event that the worker awaits. Once the worker wakes up, it reads the signal levels, calls the DMMS.R stepping function and obtains the model response (i.e., new values of BG and IG). The worker writes the obtained values to the shared memory and signalizes the event that is awaited by the daemon. Once the daemon wakes up, it reads the newly obtained values and passes them through the TCP/IP connection back to the SmartCGMS discrete model.

The network daemon has a fixed-size pool of DMMS.R worker processes. Suppose all the workers are in use and performing a simulation—in this case, any new incoming session is held in a waiting state until a worker becomes available upon finishing its previous simulation.

Currently, the daemon supports DMMS.R workers only. Nonetheless, we implemented the suite in a modular way. To support more models, development of a new daemon module only is required. The SmartCGMS discrete model can request a different model by signaling a different GUID during session establishment.

### 2.3. Results Analysis

Table *T* presents an empirical, discrete space. To identify safe regions, we aim to obtain a plausible approximation using a mathematical model. The model is then subject to risk analysis.

For the patient to validate his/her insulin pump settings, e.g., when starting with SAP therapy, we need to be able to process his/her inputs on a personalized model. Model and controller personalization is beyond the scope of this paper. Once we have a personalized model, the patient inputs his/her preferred parameter values obtained from previous successful treatment periods. Then, the suite processes the inputs and evaluates the expected treatment goodness using this parameter set. Moreover, it can suggest better insulin-pump settings (i.e., controller parameters), as the discrete space approximation will be known at this step.

The controller verification process is based on the approximation function analysis. We can perform risk analysis by identifying safe regions by obtaining a continuous function from the discrete space approximation.

We used two approaches to obtain the approximation: a least-squares approach on a manually selected binomial sum model and a decision tree.

Here, the devised model and decision tree assume only two-dimensional inputs, i.e., n=2.

#### 2.3.1. Binomial Sum Model

To obtain a feasible fit in terms of stability and simplicity, we chose to use a model obtained as a sum of binomials [33]:(3)$$z_{fit}\left(x,y\right)=\sum_{i=0}^{p}{\left(A_{i}x+B_{i}y\right)}^{i}$$

Here, *p* denotes the highest binomial order and $A_{i}$ and $B_{i}$ indicate binomial coefficients. This term further expands to
(4)$$z_{fit}\left(x,y\right)=c_{0,0}+c_{1,0}x+c_{0,1}y+c_{2,0}x^{2}+c_{1,1}xy+c_{0,2}y^{2}+\ldots$$

Here, we substitute the $A_{i}$ and $B_{i}$ parameters with $c_{m,n}$, where m=0,…,p and n=p−m for each expanded binomial. Then, we perform the least-squares method to find the $c_{m,n}$ parameters.

After a preliminary tuning phase for the degree *p*, we chose p=5 for both (two-parametric) controllers, as this has been demonstrated to be the sufficient binomial order for a given problem without exhibiting any adverse fluctuations between the approximated points.

#### 2.3.2. Decision Tree

A decision tree (DT) [34,35] is a supervised learning technique, adopting the shape of an acyclic graph that can be used to make predictions, mainly used for classification and regression tasks. This latter case provides a continuous, but non-differentiable, approximation of the control goodness function. In each tree node, a specific feature is examined and, based on the outcome, a branching takes place; at the end of a root-to-leaf path, a specific value is found in the leaf. More specifically, DT is a binary tree in which the leaves represent output values and every node that is not a leaf represents a conditional. The conditional is a simple expression in the form of <parameter> <op> <value>, where <parameter> is one of the controller parameters, <op> is the comparison operator <=, and <value> is a decision boundary value.

Every non-leaf node has exactly two child nodes connected with edges representing the conditional being evaluated as true or false. Let us refer to these nodes as *left* and *right*, for positive and negative conditional outcomes, respectively.

The decision walk starts in the root node, evaluates all given conditionals, and descends through the respective edges until it reaches the leaf node. Then, it emits the approximated output as the value stored in that node. It is worth noting that this algorithm is interpretable as trees that can be visualized.

For the implementation, we used the scikit-learn package [36] with the squared error as the function to measure the quality of a split.

We presume this approach to be very efficient for model-based controllers, such as the LGS. We expect that the evaluation of rule-based controllers will be effective only when the rules are non-contradictory, at least within close regions. This implies that the approximated space is differentiable or exhibits minimum instability in non-differentiable regions.

### 2.4. Evaluation Scenario

We need to evaluate the controllers in a scenario that yields valid and correct results in terms of certification. DMMS.R is distributed with a pre-defined scenario approved to perform in silico controller testing. A virtual patient ingests 40, 50 and 70 g of carbohydrates at 7:00 a.m., 12:00 p.m. and 7:00 p.m., respectively. We introduced no exercise and no additional event to a scenario already approved by the FDA.

To perform a fair comparison, we did not introduce any meal boluses, as the OpenAPS/*oref0* contains a mechanism to dose additional insulin to compensate for meal glucose. If we dosed additional meal boluses based on DMMS.R virtual-patient characteristics, LGS would outperform *oref0* by the principle.

DMMS.R contains diabetes type-1, type-2, and prediabetes populations. Both LGS and OpenAPS/*oref0* were originally designed for diabetes type-1. The FDA approved the 40–50–70 g scenario for diabetes type-1. Therefore, we limited this study to diabetes type-1 only. Specifically, we used child, adolescent and adult populations of diabetes type-1 patients.

### 2.5. Controller Confidence

We propose a novel metric set to reflect controller confidence. The controller must keep the IG as close to the target IG as possible. Based on knowledge of the whole parameter space, controller parameters for a single patient should be set to the values corresponding to the best outcome. To demonstrate best possible controller safety and predictability, the best outcome should be a global extremum (safety) with a maximum continuous region (predictability) around the extremum in the Cartesian product.

We do not expect the controller to exhibit ideal properties. Instead, we envisage it to have multiple extrema, preferably close to each other. The first proposed metric is the *local extrema ratio*. At first, we identify a set of local extrema $E_{l}$ from the table *T*. Then we divide it by a total count (i.e., simulation runs) of elements in the Cartesian product to obtain the metric value:(5)$$C_{l}=\frac{|E_{l}|}{\left|T\right|}$$

Using the local extrema ratio, we assess the stability and control transparency of the controller on a given patient. We deem a significant occurrence of local extrema as a warning that the controller may not exhibit predictable outcomes when a patient fine-tunes its parameters. Therefore, a lower local extrema ratio is better.

The second proposed metric is the *global extremum spread*. We identify a global extremum $x_{0}$ from the table *T*. Then we perform a breadth-first search through all the neighboring fields of the table *T* that exhibit a metric value higher than the 80th percentile of all the obtained values. We denote a set of such fields as *S*. This set represents a continuous region around the global extremum. Eventually, we count all the traversed Cartesian-product elements and divide this number by the total count of elements in the product:(6)$$C_{s}=\frac{\left|S\right|}{\left|T\right|}$$

The global-extremum-spread metric is an indicator of the global extremum region’s significance. As we chose the 80th percentile threshold, the ideal outcome for a single local extremum is 0.2, as we would expect all the relevant values to be reachable. The lower the metric value, the smaller the global extremum region; therefore, the best control parameters are more isolated from the other feasible parameters.

## 3. Results

We analyzed all the patients from a cohort available in the basic DMMS.R distribution. This cohort consisted of 10 patients for each of the three categories—adult, adolescent, and children. We did not consider the average subject because its parameters are obtained by calculating an average value from all patients. As the DMMS.R simulator provides no additional patient information, we provide no table to summarize gender, weight, age, or others.

### 3.1. Evaluation Table

As an example, we demonstrate the evaluation results on the adult#001 patient. The evaluation led to very similar results for other patients.

Figure 3 depicts the evaluation phase results as a heat map with gradient interpolation for both LGS (Figure 3a) and OpenAPS/*oref0* (Figure 3b). The green areas represent efficient, most likely safe, regions where the controller is safe for the patient. Yellow regions represent potentially unsafe parameter sets as the controller may exhibit unstable behavior. Red regions represent unsafe, potentially dangerous, regions where the control capabilities may be life-threatening. We rendered the figures using the Matlab^®^ environment.

### 3.2. Approximation

We approximated the discrete space obtained during the evaluation phase by the two models, i.e., the binomial sum model and the decision tree. For each subject, the dataset has been divided into a training and a testing set by randomly selecting the items with a split percentage for the testing set equal to 33%. The degree *p* for the binomial sum model has been set to five, while the number of maximum leaf nodes for the decision tree has been set to 100.

Table 3 reports the results for both algorithms. It is evident that the root mean squared error (RMSE) provided by the binomial sum model is always worse than that exhibited by the decision tree for both controllers. Figure 4 shows the results for both the algorithms on the subject adult#001 as an example. Table 4 reports the values of the coefficients obtained by the least-squares method when the degree *p* varies from one up to five, while Figure 5 shows the tree obtained by the decision-tree technique for the same subject when the OpenAPS/*oref0* controller is considered.

### 3.3. Controller Confidence

Table 5 shows the results of the *local extrema ratio*
$C_{l}$ and *global extrema spread ratio*
$C_{s}$ for both controllers and all the virtual patients.

Table 6 reports the average values and standard deviations for every controller and patient category. Figure 6a,b illustrate mean values of the local extrema ratio and global extrema spread ratio, respectively.

## 4. Discussion

The obtained results provide insight into the controller’s capabilities. The evaluation phase yields a discrete space. By observing the respective figures, we can identify feasible regions of control. A visible green region in the LGS controller figure spans almost the whole row of optimal basal insulin rates. This matches the expectations, as the LGS merely suspends the infusion when a hypoglycemic episode is imminent. It does not attempt to set a different basal insulin delivery rate to mitigate hyperglycemia. Opposing that, the *oref0* controller features a set of rules that modulate the basal insulin delivery rate in discrete steps, using the default rate only as a reference value. Its figure contains several fuzzy regions that are difficult to interpret. The *oref0* controller may contain contradictory rules causing the sudden jumps in the obtained metric values. Therefore, we can observe that *oref0* is associated with increased local extrema. A global extremum is located in a region around a default basal rate of 2.25U/hr and a sensitivity factor of 0.3.

The controller confidence metric values agree with the above findings. Regarding the local extrema ratio, $\bar{C_{l}}$, the values are significantly greater for *oref0* than for the LGS controller, i.e., the *oref0* evaluation table contains a greater number of local extrema than the LGS table. This confirms the empirically observed facts. With respect to the global extremum spread metric, $\bar{C_{s}}$, the results highlight an interesting finding—the LGS tends to exhibit fewer (less than *oref0* has) well-performing regions spread over the whole sampling region. In contrast, the *oref0* controller often shows a strong global extremum surrounded by a greater (than LGS) number of good-performing parameter sets. This means that the *oref0* controller tends to have a single large well-performing region and, according to the local extremum ratio, a substantial number of smaller regions of acceptable performance.

With respect to the approximation, the results exhibited by the decision tree outperform those of the binomial sum model. The decision tree also has an advantage over the binomial model—safe control regions can be obtained directly from the rules, as the rules establish hyper-rectangular (rectangular with n=2) regions with a metric value less than a given value. Nevertheless, the experimental results show that the loss in performance between the training and testing sets is about an order of magnitude for the decision tree, while, for the least-squared method, the performance is the same. On the other hand, complex decision trees tend to overfit and do not generalize well to new data. This problem can be avoided by using a random forest approach [37], which consists of multiple trees designed to increase the performance. Nevertheless, in this case, the interpretability of the trees is much more complex.

The use of an in silico model is important to obtain realistic results. Therefore, we chose the FDA-approved DMMS.R metabolic simulator. This simulator already accounts for various fluid flow factors, metabolic disturbances, CGM sensor signal noise, physical-activity effects, and others. Nevertheless, research and development of new metabolic simulators is an ongoing process. For example, we proposed a new approach to metabolic modelling [38]. With a different, possibly new simulator, we can use the SmartCGMS ability to change the used in silico simulator to another one as easily as only modifying the experimental setup [19].

## 5. Conclusions and Future Work

We proposed a method for controller evaluation that integrates an FDA-approved human metabolism simulator DMMS.R, the SmartCGMS software stack, and a set of tools for distributed computing to reduce the execution time for evaluation. The SmartCGMS is a software framework designed to run on low-power devices. Hence, when used for controller evaluation, it can be directly deployed to an insulin pump without changing the code base. The distributed approach is required to speed up the evaluation, as many parameter combinations are exploited to obtain the approximation.

The second phase of the evaluation is to obtain an approximation of the discrete space calculated during the first phase. The approximation serves as the function of control goodness. We propose to use it in two cases: the controller pre-verification step and parameter validation for patient use.

Several aspects of this study generate a need for further research and improvements. A limitation of the current approach is the problem dimension. We only evaluated two controllers, each with two parameters. As modern algorithms often use more parameters to improve fine-tuning for a specific patient, further research on the method should be undertaken on problems with more significant dimensions. For example, several successful approaches use PID-based controllers with five or more parameters. The framework itself is sufficiently generic to address such problems.

A further potential subject of future research relates to the pre-verification steps and risk analysis of the obtained control goodness function. As we can obtain a continuous function describing the control capabilities, we can use methods of mathematical analysis to identify safe regions. This is why we investigated both binomial sum and decision-tree approximations.

For reasons given in Section 2.4, this study concerns patients with diabetes type-1. In future work, we plan to investigate other types of diabetes and patient populations. This will help us to determine outcomes for patients with different insulin responses.

## Figures and Tables

**Figure 1 sensors-22-09445-f001:**
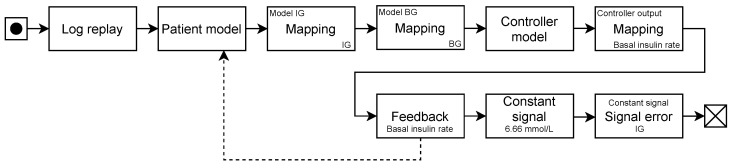
A filter chain configured for in silico testing of a selected controller.

**Figure 2 sensors-22-09445-f002:**
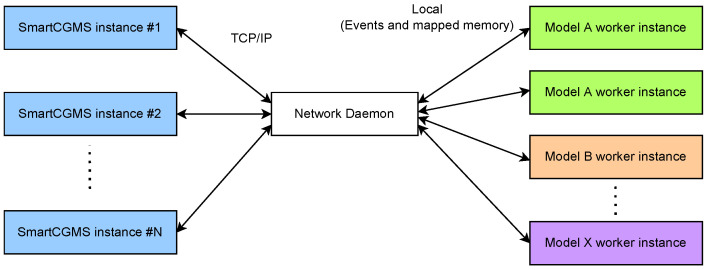
Scheme of in silico evaluation suite.

**Figure 3 sensors-22-09445-f003:**
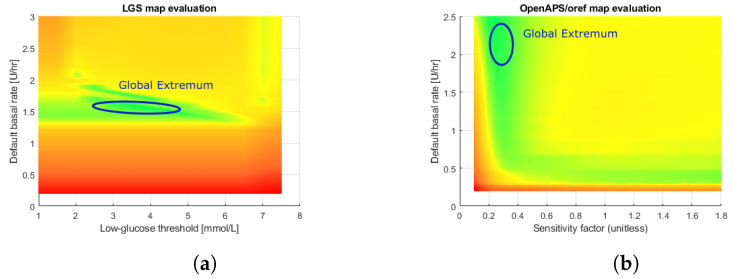
An evaluation map of two controllers on adult#001 subject of DMMS.R. Blue ellipses denote global extrema for each controller, while the rest of the green areas contain the local extrema. (**a**) LGS controller (**b**) OpenAPS/*oref0* controller.

**Figure 4 sensors-22-09445-f004:**
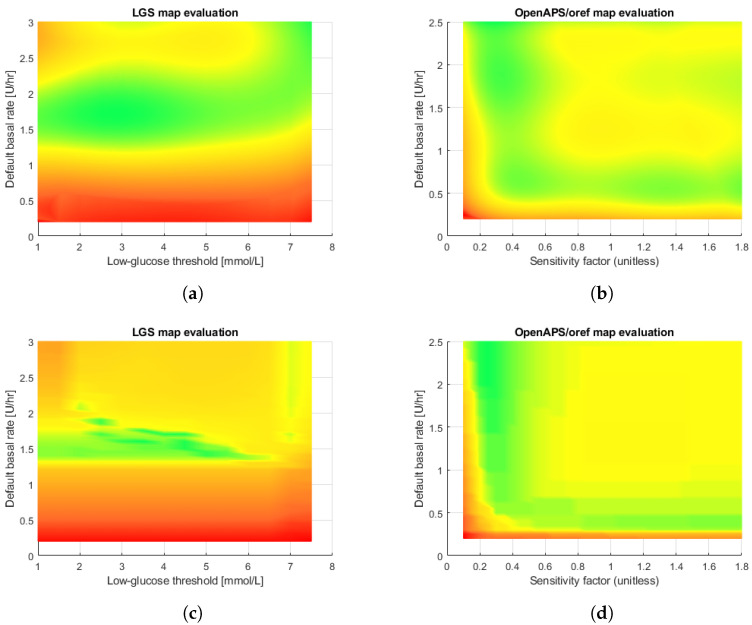
Obtained approximations by the the binomial sum model with degree 5 (top panes) and decision tree (bottom panes) on adult#001 subject of DMMS.R. (**a**) LGS controller. (**b**) OpenAPS/*oref0* controller. (**c**) LGS controller. (**d**) OpenAPS/*oref0* controller.

**Figure 5 sensors-22-09445-f005:**
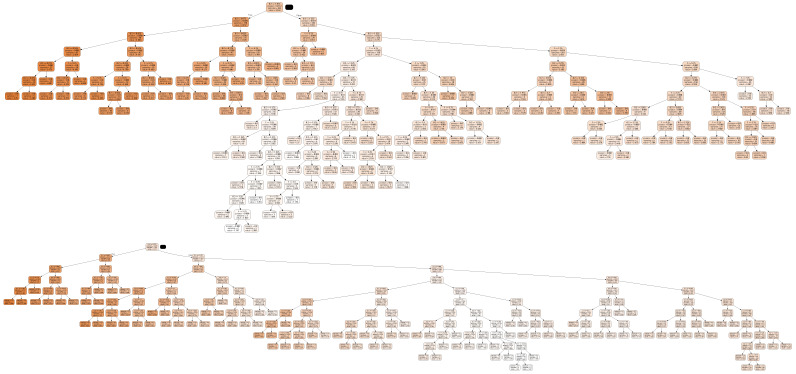
Decision tree for both the controllers on adult#001 subject of DMMS.R.

**Figure 6 sensors-22-09445-f006:**
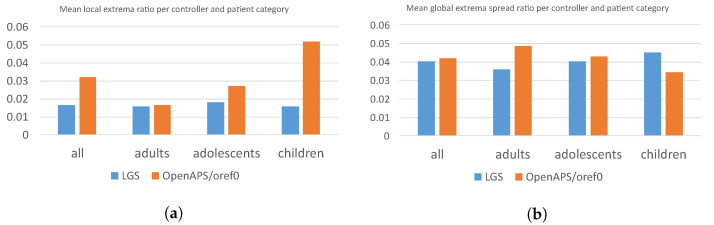
Graphs of confidence metric average values for both the controllers and each respective category. (**a**) Local extrema ratio average $\bar{C_{l}}$ (**b**) Global extremum spread ratio means $\bar{C_{s}}$.

**Table 1 sensors-22-09445-t001:** Comparison table of existing in silico evaluation suites and methods.

	S2008 [30]	S2013 [31]	S2017 [18]	T1DMS [17]	DMMS.R [17] (Stand-Alone)	SimGlucose [32]	Proposed Approach (SmartCGMS [19] with DMMS.R [17])
Type	Simulator	Simulator	Simulator	Suite ^1^	Suite ^1^	Suite ^1^	Suite ^1^
Licensing	Described as equations in published studies			Proprietary	Proprietary	MIT	Mixed ^2^
Distribution	None	None	None	Binary	Binary	Source	Both
Regulatory-body approval	Yes	Yes	Yes	Yes	Yes	No	Partial ^3^
T1D patient cohort	Undisclosed	Undisclosed	Undisclosed	30	30	30	30
Subc. insulin	Yes	Yes	Yes	Yes	Yes	Yes	Yes
Subc. glucagon	No	Yes	Yes	No	Yes	No	Yes
Physical activity	No	No	Yes	No	Yes	No	Yes
Diabetes types	T1D	T1D	T1D, T2D, Pre	T1D	T1D, T2D, Pre	T1D	T1D, T2D, Pre
Many patient evaluation	No	No	No	Yes	Yes	Partial	Yes
Many controller evaluation	No	No	No	No	No	No	Yes

^1^ Suite comprises a simulator and an open standardized way to implement a custom regulator. ^2^ SmartCGMS is distributed under a GPLv3 license, but licensing options may vary depending on the simulator used. ^3^ The proposed suite uses DMMS.R that is approved by the FDA, but SmartCGMS is not yet approved.

**Table 2 sensors-22-09445-t002:** Parameter bounds and increments for evaluated controllers.

LGS	*b*	Θ	*oref0*	*b*	*s*
*L*	0.2	1.0	*L*	0.2	0.1
*U*	3.0	7.5	*U*	2.5	1.8
*S*	0.05	0.5	*S*	0.05	0.05

**Table 3 sensors-22-09445-t003:** Results of the binomial sum (BS) model and the decision tree (DT) in terms of average RMSE (the values in brackets are the standard deviations).

		LGS			
		All	Child	Adolescent	Adult
BS	Train	0.0940 (0.0743)	0.0536 (0.0154)	0.0485 (0.0052)	0.1798 (0.0725)
	Test	0.0997 (0.0806)	0.0569 (0.0176)	0.0505 (0.0066)	0.1916 (0.0802)
DT	Train	0.0004 (0.0006)	0.0001 (0.0000)	0.0001 (0.0000)	0.0010 (0.0006)
	Test	0.0068 (0.0113)	0.0022 (0.0025)	0.0006 (0.0004)	0.0177 (0.0140)
		**OpenAPS/** * **oref0** *			
		**All**	**Child**	**Adolescent**	Adult
BS	Train	0.1811 (0.0736)	0.1602 (0.0654)	0.2217 (0.0380)	0.2217 (0.0380)
	Test	0.1839 (0.0735)	0.1595 (0.0633)	0.2260 (0.0344)	0.2260 (0.0344)
DT	Train	0.0008 (0.0007)	0.0005 (0.0004)	0.0010 (0.0010)	0.0009 (0.0005)
	Test	0.0100 (0.0067)	0.0107 (0.0082)	0.0125 (0.0047)	0.0067 (0.0053)

**Table 4 sensors-22-09445-t004:** Obtained coefficients by the least-squares method for both the controllers on adult#001 subject of DMMS.R.

Degree *p*	Binomial Sum Coefficients					
	LGS					
1	2.29$\times 10^{1}$	−2.28$\times 10^{1}$				
2	5.04$\times 10^{1}$	4.32$\times 10^{0}$	3.62$\times 10^{1}$			
3	−5.31$\times 10^{1}$	−2.68$\times 10^{0}$	4.70$\times 10^{-1}$	−2.77$\times 10^{1}$		
4	2.61$\times 10^{1}$	1.54$\times 10^{0}$	−7.39$\times 10^{-1}$	1.51$\times 10^{-1}$	9.88$\times 10^{0}$	
5	−4.82$\times 10^{0}$	−4.51$\times 10^{-1}$	3.02$\times 10^{-1}$	2.36$\times 10^{-2}$	-8.61$\times 10^{-3}$	−1.33$\times 10^{0}$
	OpenAPS/*oref*					
1	9.38$\times 10^{0}$	2.20$\times 10^{0}$				
2	−1.58$\times 10^{1}$	−2.26$\times 10^{0}$	−9.67$\times 10^{-1}$			
3	9.01$\times 10^{0}$	1.57$\times 10^{0}$	2.98$\times 10^{-1}$	2.34$\times 10^{-1}$		
4	−1.98$\times 10^{0}$	−5.97$\times 10^{-1}$	−6.63$\times 10^{-2}$	−1.77$\times 10^{-2}$	−2.89$\times 10^{-2}$	
5	1.39$\times 10^{-1}$	6.92$\times 10^{-2}$	1.98$\times 10^{-2}$	−3.31$\times 10^{-3}$	7.90$\times 10^{-4}$	1.38$\times 10^{-3}$

**Table 5 sensors-22-09445-t005:** Controller confidence metrics.

	LGS		OpenAPS/*oref0*	
	$C_{l}$	$C_{s}$	$C_{l}$	$C_{s}$
child#001	1.503%	4.637%	5.106%	4.985%
child#002	1.629%	4.762%	6.079%	3.647%
child#003	1.378%	4.386%	5.167%	2.796%
child#004	1.003%	4.010%	2.675%	4.742%
child#005	1.378%	4.887%	5.532%	4.985%
child#006	1.629%	4.511%	6.687%	4.012%
child#007	1.504%	4.887%	5.228%	4.985%
child#008	2.381%	3.759%	4.377%	0.729%
child#009	2.256%	4.261%	4.559%	0.243%
child#010	1.253%	4.887%	6.444%	3.100%
adolescent#001	1.504%	1.880%	3.830%	4.985%
adolescent#002	2.005%	4.511%	4.985%	4.985%
adolescent#003	1.378%	4.887%	1.763%	3.283%
adolescent#004	1.504%	1.880%	4.498%	2.371%
adolescent#005	3.008%	4.511%	1.763%	4.802%
adolescent#006	1.880%	3.509%	1.581%	4.985%
adolescent#007	1.629%	4.887%	4.012%	4.985%
adolescent#008	1.504%	4.887%	1.155%	4.985%
adolescent#009	2.130%	4.887%	1.459%	2.553%
adolescent#010	1.629%	4.386%	2.310%	4.985%
adult#001	2.757%	4.261%	1.398%	4.863%
adult#002	0.627%	4.887%	3.100%	3.891%
adult#003	1.754%	4.887%	1.155%	4.985%
adult#004	1.754%	2.882%	0.669%	4.985%
adult#005	1.128%	1.003%	0.972%	4.985%
adult#006	1.253%	2.882%	1.824%	4.924%
adult#007	1.378%	4.887%	2.614%	4.985%
adult#008	3.133%	4.135%	2.310%	4.924%
adult#009	1.128%	4.887%	1.459%	4.985%
adult#010	0.877%	1.253%	1.094%	4.985%

**Table 6 sensors-22-09445-t006:** Average (standard deviation) values of obtained controller-confidence metrics per patient category.

	LGS		OpenAPS/*oref0*	
	$\bar{C_{l}}$	$\bar{C_{s}}$	$\bar{C_{l}}$	$\bar{C_{s}}$
children	1.591% (0.404%)	4.499% (0.374%)	5.185% (1.101%)	3.423% (1.653%)
adolesc.	1.817% (0.447%)	4.022% (1.142%)	2.736% (1.361%)	4.298% (1.043%)
adults	1.579% (0.763%)	3.596% (1.437%)	1.660% (0.747%)	4.851% (0.323%)
all	1.662% (0.575%)	4.039% (1.142%)	3.194% (1.840%)	4.188% (1.286%)

## Data Availability

Data were generated using the DMMS.R in silico patient cohort provided in its basic distribution, namely patients child#001 through child#010, adolescent#001 through adolescent#010 and adult#001 through adult#010.

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
