# Peer review of "Distributed Assessment of Virtual Insulin-Pump Settings Using SmartCGMS and DMMS.R for Diabetes Treatment"

_sensors, 2022, doi:10.3390/s22239445_

Round 1

Reviewer 1 Report

The article is Interesting and useful. It can be accepted with fellowing modification.

1.      Title of article should be change to appropriate aligned with study.

2.      There are various fluid flow factors, these should be added to model or simulator.

3.      What is the different between current study and existing model?

4.      Bench mark table should be added by giving the compression of current study with literature by giving its pros and cons.

Reviewer 2 Report

Due to the significant impact of diabetes on public health, this work has the potential to benefit diabetes patients. The insulin pump is believed one of the solutions to improve the life quality of diabetes patients who need insulin injections to fight against high blood glucose levels in a timely. 

The patents could be grouped according to different clinical categories for the simulation work. Now, the author divided the patients into adults, adolescents, and children.

Other than that, it could be better to consider the subtype diabetes type, such as type 1 and 2, which has different insulin response.

In addition, the weight, gender, and stage of the disease may also impact insulin consumption. The author can validate their simulation on those subgroups of patents, enhancing its potential in clinical transition. 

Round 2

Reviewer 2 Report

The authors responded well items by items according to the reviewers comments and suggestions. Now, this works could be published to the Sensors